# Room Temperature Synthesized TiO_2_ Nanoparticles for Two-Folds Enhanced Mechanical Properties of Unsaturated Polyester

**DOI:** 10.3390/polym15040934

**Published:** 2023-02-13

**Authors:** Muhammad Shoaib, Zeeshan Latif, Mumtaz Ali, Ahmed Al-Ghamdi, Zafar Arshad, S. Wageh

**Affiliations:** 1School of Engineering and Technology, National Textile University, Faisalabad 37610, Pakistan; 2Department of Physics, Faculty of Science, King Abdulaziz University, Jeddah 21589, Saudi Arabia

**Keywords:** room-temperature synthesis, TiO_2_ nanoparticles, mechanical properties, nano-reinforcements, polymer composites

## Abstract

Using of nano-inclusion to reinforce polymeric materials has emerged as a potential technique to achieve an upper extreme of specific strength. Despite the significant improvement of mechanical properties via nano-reinforcements, the commercial application of such nano-composites is still restricted, due to high cost and unwanted aggregation of nanoparticles in the polymer matrix. To address these issues, here we proposed a scalable and economical synthesis of TiO_2_ at low temperatures, resulting in self-dispersed nanoparticles, without any surfactant. As lower energy is consumed in the synthesis and processing of such nanoparticles, so their facile gram-scale synthesis is possible. The defect-rich surface of such nanoparticles accommodates excessive dangling bonds, serving as a center for the functional groups on the surface. Functional surface enables high dispersion stability of room temperature synthesized TiO_2_ particles. With this motivation, we optimized the processing conditions and concentration of as-synthesized nano-particles for better mechanical properties of unsaturated polyester (UP) resin. The composite structure (UP-TiO_2_) showed nearly two folds higher tensile, flexural, and impact strength, with 4% content of nanoparticles. Characterization tools show that these better mechanical properties are attributed to a strong interface and superior dispersion of nanoparticles, which facilitate better stress distribution in the composite structure. In addition, the crack generation and propagation are restricted at a much smaller scale in nanocomposites, therefore significant improvement in mechanical properties was observed.

## 1. Introduction

Polymers are reinforced with different types and sizes of fillers to eradicate the constraints of pure polymers and diversify their end applications [1]. Specifically, particulate fillers reinforce the polymer in various modes, including the enhancement of physio-mechanical characteristics, induced by the strong and large area interface [2]. Based on this, a better interface is expected for the nanomaterials, as exceptionally high surface areas can be achieved at the nanoscale [3]. Mechanical properties apart, unique functionalities can be introduced in the polymers via nano-reinforcements, for instance, antibacterial activity, UV resistance, and fire resistance [4]. Several researchers have employed nanoparticles/nanorods of various materials such as calcium carbonate, silicon dioxide, zinc oxide, and titanium dioxide (TiO_2_) to enhance the physio-mechanical properties. The inclusion of these nanofillers in polymer provides additional filler-matrix interaction sites due to the large surface area and hence improves the mechanical strength [5,6,7]. This study is an extension of the previous research; we applied the low-temperature synthesized TiO_2_ nanoparticles, unlike conventional complex processed nanoparticles. 

Importantly, nanofillers increase the creep performance, stiffness, hardness, and thermal stability of the composite matrix [7]. To improve mechanical performance, nanofillers fill cavities and lower the free volume in the matrix [8]. The interlaminar shear strength values of the nano-SiO_2_ modified-sized carbon fiber reinforced composites were raised by 9% and 14%, respectively, when compared to the unsized composites. As the sizing can improve mechanical interlocking by increasing the fiber’s surface roughness and Sizing dissolved into the matrix is possible to improve chemical interaction between the matrix and the sized fibers [9]. Prepared copper matrix composites by the in-situ generation of the nano TiC phase using CPD as the carbon source. As a result of the dislocation inhibition generated by nano-carbide and strong interface bonding between nano TiC and the Cu matrix, the 0.3 wt% CPD/Cu composite exhibits the highest strength- plastic compatibility, with an ultimate tensile strength of 385 MPa and a matching elongation of 21%. These benefits of the preparative method include stronger bonds at the interface, better dispersion, and pure chemical composition at the surface [10]. Pressureless sintering of Ti_3_C_2_T_x_ and Al powders at 650 °C for 1 h in Ar, followed by hot extrusion at 460 °C, has been used to make Ti_3_C_2_T_x_/Al composites. Ti_3_C_2_T_x_’s wettability with Al is improved by the abundance of functional groups on its surface. Ti_3_C_2_T_x_/Al composites are endowed with enhanced mechanical characteristics due to the uniform distribution of Ti_3_C_2_T_x_ and the creation of strong bonding surfaces in the composites. From a weight percent of 0.5 wt% of Ti_3_C_2_T_x_ to 3 wt%, there is a linear rise in Vickers hardness and tensile strength. When compared to pure Al, the hardness and tensile strength of the composite made up of 3 wt% Ti_3_C_2_T_x_/Al are 92% and 50% higher, respectively [11]. Partially wrapped MWCNTs (Py-PPDO)_2_-b-PEG@MWCNTs) were prepared by the synthesis of a triblock copolymer of (Py-PPDO)_2_. The incorporation of (Py-PPDO)_2_-b-PEG@MWCNTs nano-aggregates enhanced the stability and dispersion of GE in solvent and polymer matrix as a result of the interaction between MWCNTs and GE. Using GE/(Py-PPDO)_2_-bPEG@MWCNTs as hybrid nanofillers, PCL nanocomposite films showed significantly enhanced mechanical characteristics compared to pure PCL even at very low nanofiller concentration. By adding only 0.01 wt% of GE/MWCNTs, the tensile strength and elongation at break were dramatically raised by 163% and 17%, respectively. A transmission electron microscopy study of the composite films indicated that the (Py-PPDO)_2_-b-PEG@MWCNTs not only inhibited the aggregation of GE but also facilitated the transfer of loads between the polymer matrix and the nanofillers [12]. Previous studies employed nano-fillers of CaCO_3_ [5], SiO_2_ [6], ZnO [7], and TiO_2_ [8] to reinforce the polymer resins. TiO_2_ nanoparticles are extensively employed by researchers in the polymer matrix as filler because of their strong mechanical properties. In addition, TiO_2_ nanoparticles also offer an excellent refractive index, resilience against corrosion, good photocatalytic activity, and UV resistance [13]. Conventionally, TiO_2_ nanoparticles are synthesized through high-temperature reflux, solvothermal, and hydrothermal techniques. These techniques operate at high temperatures, hence are not time and energy-efficient [14]. During high-temperature processing, TiO_2_ nanoparticles tend to aggregate via Ostawrld’s ripening, resulting in larger crystals, with a lower surface area [15]. To overcome these issues, a low-temperature solution synthesis of TiO_2_ nanoparticles was effectively reported to replace the conventional synthesis process [16].

Pottier et al. synthesized TiO_2_ nanoplatelets at low temperatures by the reaction of TiCl_4_ in the presence of concentrated HCl [17]. Di Paola et al. fabricated different combinations of TiO_2_ phases including the rutile phase, binary mixed rutile and brookite phase, and tertiary mixed rutile, anatase, and brookite phases by applying the same approaches as mentioned previously [18]. Fischer reported the fabrication of low-temperature synthesized tertiary phased (rutile, anatase, and brookite) TiO_2_ nanoparticles on the polymer membrane for wastewater application [19]. Hossain et al. reported a scalable facile approach for low-temperature synthesized TiO_2_ nanoparticles, and their application in efficient perovskite solar cells [20]. Tan et al. synthesized the TiO_2_ nanoparticles using low temperature, organic-free technique and reported its photocatalytic activity for the degradation of phenol [21]. At present, researchers reported the low-temperature TiO_2_ nanoparticles in various fields including water treatment [22], solar cells [23], energy storage devices [23], and landfill leachate treatment [24]. However, no study has been reported regarding the integration of these low-temperature fabricated TiO_2_ nanoparticles in the polymer matrix to enhance the mechanical properties. As these nanoparticles have a higher functional group and dangling bond at their surface compared to conventional, so it is expected that these nanoparticles provide a better interface between nanoparticles and matrix compared to conventional. Bearing this in mind, low-temperature synthesized TiO_2_ nanoparticles were integrated with unsaturated polyester (UP) resin to improve mechanical performance.

In this study, low-temperature synthesis of TiO_2_ nanoparticles and their inclusion in UP resin to enhance the mechanical properties was carried out. The molding conditions of the matrix (including temperature, time, and pressure) and the concentration of TiO_2_ nano-inclusions were optimized. The formation of TiO_2_ particles via low-temperature solution processing was confirmed using scanning electron microscopy (SEM). Mechanical properties of the TiO_2_-reinforced polymer were tested as a response variable, such as tensile strength, flexural strength, impact strength, and hardness. In addition, the antibacterial functionality imparted by the TiO_2_ inclusion was also tested. Based on the suitable enhancement of mechanical properties and functionality via a facile process, we expect that low-temperature synthesized TiO_2_ will advance the application of nano-composites at an industrial scale.

## 2. Materials and Methodology

### 2.1. Materials

Titanium tetrachloride (TiCl_4_, 94%) and hydrogen peroxide (H_2_O_2_, 35%) were procured from Sigma Aldrich, (St. Louis, MO, USA). Unsaturated Polyester resin, methyl ethyl ketone peroxide (MEKP), and cobalt were purchased from the Nimir group of industries, in Lahore, Pakistan. Ammonium hydroxide (NH_4_OH, 88%), used as a catalyst for the TiO_2_ synthesis, was purchased from Sigma Aldrich. All chemicals used were of analytical grade. Deionized water and washed glassware were used throughout the experiment.

### 2.2. Synthesis of TiO_2_ Nanocrystals

TiO_2_ nanocrystals were synthesized by a room-temperature sol-gel method employing TiCl_4_ as shown in Figure 1. Room temperature synthesis of Titania nanoparticles is energy efficient economical and scalable for large-scale synthesis, essentially required for composite applications. Low temperature and ambient condition synthesis is a feasible process to be used for nano-materials synthesis, however, its compatibility with polymeric resins is not studied yet. The mechanism of room-temperature synthesis of nanoparticles follows four steps; hydrolysis, polycondensation, stabilization, and finally separation/drying. Initially, 5.5 mL TiCl_4_ was added into 200 mL distilled water, dropwise; which leads to the formation of Titanium oxychloride (TiOCl_2_). This intermediate is unstable; hence it decomposes with the release of hydrogen chloride (HCl). This reaction is highly exothermic and care must be taken to keep the temperature low. To achieve safe reaction conditions, the solution bath was placed in an ice bath at −5 °C. It was allowed to stir for 30 min and then 2 mL of ammonium hydroxide (NH_4_OH) was added slowly for hydroxide gel formation. The pH was maintained at higher than 2 and TiO_2_ nanocrystals were formed, indicated by the solution color, which was changed to white. The prepared particles were then centrifuged at 4000 rpm for 15 min and finally washed with distilled water for the removal of NH_4_Cl. We received 2.98 g TiO_2_ particles as the product.

### 2.3. Fabrication Methodology of Composites

Considering the set objectives, the effect of parameters like (curing temperature, concentration of TiO_2_, and residence time) were studied. The sampling was carried out using the compression moulding technique, using initiator MEKP (1%), cobalt as accelerator (2.5%) & UP resin. The effect of the concentration of reinforcing agent TiO_2_ is by varying its quantity, i.e., 0%, 2%, 4%, and 6%.

Polyester resin mass was determined from the volume of the cast, and the filler material mass was added to 3% of the total resin. The filler TiO_2_ particles and the resin were mixed at room temperature, using a mechanical stirrer at 500 rpm for 30 min. For degassing the TiO_2_-UP composite, the particle mixture was placed under a vacuum for 15 min. 1% weight fraction MEKP initiator was added and stirred till homogeneity. Afterward, the accelerator which was 2.5% of the resin was mixed gently to avoid bubbling in the mixture. This mixture of UP, TiO_2_, MEKP, and cobalt was poured from one corner into the mold to avoid bubble formation which causes cast damage and uniform pouring is continued until the mold was filled to the required level [25]. Mould was placed on the electrical vibrator to remove any residue bubbles. Curing temperature, residence time, and the reinforcing agent was varied to study its effect on mechanical properties [26]. The precuring of the composite was carried out at room temperature for 12 h, afterward post-curing at 120 °C for 2 h [26].

### 2.4. Characterizations

The surface morphology of the TiO_2_ nanoparticles was characterized using a scanning electron microscope (SEM-JEOL JSM-7600F) Tokyo, Japan; working on 40 KV. Images were taken at different resolutions to detect the dispersion and the exact size of synthesized titania nanoparticles. Structural and crystallographic characterizations were determined via X-ray diffraction (XRD) (Bruker axis D8 Advance diffractometer, Berlin, Germany) at 45 kV. It scanned samples over the 2θ range of 10–60°, at a scanning rate of 0.02 s^−1^. The radiation source was Cu Kλ (λ = 1.5406 Å) at 30 mA. Fourier-transform infrared spectroscopy (FTIR) spectrum was analyzed to detect functional groups at a resolution of 4 cm^−1^. Samples were scanned between 400–4000 cm^−1^ using Perkin Elmer Spectrum 400, Tokyo, Japan in attenuated total reflectance (ATR) mode. A flexural test was used to measure composite flexibility, approximated from the force necessary to bend a beam under a specific loading condition. Typically, a three-point bending condition according to ASTM D790 was followed to measure flexural strength. Tensile strength: a key mechanical property, was tested according to ASTM D638-02a. Flexural and tensile strength tests were performed for the hybrid composite’s strength using UTM TIRA test 2810, equipped with a load cell of 10 kN. The impact strength was tested to understand the deformation behavior under a sudden force, which was tested according to ASTM (D 6110). Impact energy was calculated by using a pendulum impact tester. The hardness test was performed according to ASTM D785-03, using a Brinell hardness machine (Gunt Gerätebau GmbH Fahrenberg 14, Postfach 1125, 22885 Barsbüttel Germany). Qualitative antibacterial testing was performed using *Escherichia coli* (*E. coli*) and *Staphylococcus aureus* (*S. aureus*) having test standard AATCC 147. A single colony of *S. aureus* bacteria and samples were added to the broth, and it was incubated at 37 °C for 24 h.

## 3. Results and Discussion

The resin/matrix is an integral part of composites, being a continuous phase, it determines the stress distribution in the whole structure. For instance, fiber-reinforced polymer composites are currently emerging as an alternative to the existing metallic parts, thanks to their high specific strength [27,28]. In such composites, the weaker component is the polymer matrix, hence the overall performance of the composite is determined by the resin. Unsaturated polyester (UP) resin is one of the most commonly used thermoset resins, being used for composite parts. To enhance the mechanical properties of the resin, nano reinforcements are the best candidate, as they stop the crack generation and propagation at a much smaller scale. This work is the extension of the same idea, where scalable synthesis was utilized for nanoparticle synthesis at room temperature for the nano-reinforcement of UP resin.

Low-temperature synthesis of nano-particles is a self-driven exothermic process and is relatively faster compared to the conventional hydrothermal process. This process enables the scalable synthesis of TiO_2_ nanoparticles with an economical process for composite applications. Besides scalability and low cost, TiO_2_ nanoparticles synthesized at low temperatures offer higher dangling bonds on the surface. Such a functional surface creates stronger links between the matrix and nanoparticles in the composite. These linkages provide a better interface and suppress the agglomeration of nanoparticles in resin. Consequently, better performance, with a lower cost is expected for the room-temperature synthesized nano-particles.

### 3.1. Characterizations of Nano-Particles and Composites

To confirm the successful synthesis of TiO_2_ nanoparticles, SEM was performed for the TiO_2_ powder, as shown in Figure 2. As nanoparticles are processed and synthesized at low temperatures, morphology is highly uniform and controlled. As at low temperatures, precursors are not randomly aggregated due to slow reaction kinetics, so particle size distribution is not widely distributed. At slower reaction rates, the size of particles and growth were well controlled. TiO_2_ particles showed a uniform spherical shape, with an average size distribution of 50–80 nm. The formation of solid particles with even diameters of small size induces superior mechanical performances in the composites [29]. Qualitatively, the synthesis conditions and precursors’ concentration were initially optimized to achieve a higher production rate. With the final conditions, the production yield of nano-particles was above 95%, concerning the mass of precursors added, which is higher than the conventional high-temperature synthesis [30].

XRD analysis was performed to analyze the crystal structure of UP resin, TiO_2_,nanoparticles and TiO_2_-UP nanocomposite, as shown in Figure 3. UP resin revealed two crystalline peaks formed at 2θ = 21.5° and 43°, indicating an inter-planar distance of 3.90 and 3.44 Å, as shown in Figure 3a. The broader peak at 21.5° arises from the major axis of crystalline domains formed in UP resin. The low intensity and broad nature of these UP resin peaks are associated with the small size and larger interplanar spacing of crystalline domains, as commonly observed in polymers [26]. XRD spectra TiO_2_ nanoparticles are shown in Figure 3a. XRD showed the anatase phase of TiO_2_, with peaks positioned at angles (2θ) of 27.5°, 36.8°, 37.02°, 41.89°, 44.06°, 54.7°, and 56.6°, associated with the planes (101), (110), (103), (004), (112), (105), and (211), respectively. These referred peaks are corresponding with anatase TiO_2_, as evaluated from JCP (21-1272) [31]. TiO_2_ inclusions in UP resin decreased the peak intensity and broadened the intrinsic UP resin peaks [26]. This could be related to the mutual interactions of TiO_2_ and UP resin [32]. The peaks at 27.5°, 36.9°, 37.2°, 41.5°, 44.5°, 54.3°, and 56.75° are characteristic peaks of TiO_2_ nanoparticles added to the resin. The crystal plane associated with each peak is labeled in Figure 3a, where the sharp and high-intensity peaks of TiO_2_ are due to the larger size of crystalline domains formed in nano-particles. It is important to note that the intrinsic peaks of TiO_2_ are not shifted after their embedment in UP, which confirms that TiO_2_ retains its original structure in the composite structure.

Fourier transform infrared analysis was performed to probe the variation in functional groups of UP resin and composite, as shown in Figure 3b. Absorption spectra were recorded between 4000 cm^−1^ to 600 cm^−1^. The presence of Titania in the polyester resin matrix leads to a partial chemical crosslinking of the TiO_2_ surface with UP resin. Functional groups related to each FTIR peak are labeled in Figure 3b, which are related to oxygenated functional groups and carbon backbone. The molecular vibrations showed a slight increase in peak intensity after titania doping, which is due to the participation of pendant functional groups on the TiO_2_ surface. This increases carbon crosslinking with nano-crystals and induces better mechanical strength in composites. Specifically, the intensity of -OH groups is significantly improved in composite, which is related to dangling bonds on the interface of TiO_2_ [33]. This chemical crosslinking of polymer chains on nan-crystals also leads to a decreased degree of crystallinity of the matrix, as observed in XRD results (Figure 3b). Mainly, the polar nature of -OH bonds (highlighted in the range of 3250 to 3600 cm^−1^), offers a strong affinity for the adhesion of polymer chains on the surface [33]. Furthermore, oxygenated bonds at 1750 cm^−1^ (C=O) and 1300 cm^−1^ (C-O-C) were slightly shifted to a lower frequency for the composite structure, suggesting the restricted bond stretching in the presence of TiO_2_ nanocrystals [34].

### 3.2. Mechanical Testing of TiO_2_ Reinforced Composites

Composites are commonly prone to different types of loads, therefore their mechanical properties are crucial in determining their performance [30,32,33,34,35,36,37,38,39]. Here we tested the mechanical properties such as tensile strength, flexural strength, Brinell hardness, and impact strength, for testing the optimization of processing conditions. These mechanical responses were checked over the operating parameters of the compression moulding machine, such as curing temperature, compression pressure, residence time, and the concentration of the reinforcing TiO_2_ nanoparticles.

#### 3.2.1. Effect of Curing Temperature

Curing is the chemical crosslinking of polymer chains to toughen or harden the polymer material. This process is observed in thermosetting polymers, where curing results in the transformation of a liquid resin into a solid structure [28]. In this work, the mechanical properties of the composites were tested at three different curing temperatures: 60, 80, and 100 °C. It is important to note that other parameters, like concentration of nano-particles (2%), pressure (80 bar), and time of curing (30 min) were kept constant for all samples. Mechanical response at different temperatures showed a significant difference, for instance, a notably better tensile strength of the composites cured at 80 °C is obvious (Figure 4a). The better tensile response is directly related to better crosslinking and toughness of the composite. While composite cured at 100 °C showed less tensile strength due to degradation of the matrix and over-crosslinking induced brittleness. TiO_2_-UP composite fracture behavior is also discussed in the next section. In contrast, the sample cured at 60 °C showed lower performance due to incomplete crosslinking of the matrix. The tensile modulus is the initial slope of the tensile curve, which indirectly relates to the stiffness of the material. The composites cured at 60, 80, and 100 °C showed tensile moduli of 14.5, 6.6, and 14.7 GPa, respectively (Figure 4b). Similarly, the flexural modulus which denotes the resistance of a material against bending forces showed a similar trend at different curing temperatures (Figure 4c). The flexural modulus of 1.6, 2.9, and 2.7 GPa at 60, 80, and 100 °C, respectively. Also, the hardness of a material is the resistance to localized plastic deformation, composites showed Brinell hardness of 74, 78, and 72 at curing temperatures of 60, 80, and 100 °C, respectively (Figure 4d). For the above given mechanical characterizations, the composites’ superior performance was observed at 80 °C curing temperature. Although, composite cured at 60 °C showed a poor performance against low-velocity mechanical deformations is due to less stiffness. However, the impact strength is measure of energy endured under abrupt loads was relatively higher for the 60 °C cured composites (Figure 4e). Composites showed impact strength of 27.4, 25.8, and 23.2 J at 60, 80, and 100 °C, respectively. Composite cured at 60 °C showed relatively higher energy absorbance due to less compactness of the campsites and less stiffness than others. The ductile nature of the composite cured at 60 °C renders higher impact tolerance.

In summary, we can conclude that the curing temperature of 80 °C offers better mechanical properties. Hence other processing parameters of the composite were optimized while keeping the curing temperature at 80 °C.

#### 3.2.2. Effect of Compression Pressure

Compression moulding pressure influences the density, crystallinity, melting flow index, ultimate tensile strength, and Young’s modulus. In this work, the mechanical properties of the composites were tested against three different compression pressures 80, 100, and 120 bar. The curing temperature of 80 °C and the concentration of TiO_2_ particles (2%) were kept constant for all samples. Stress-strain response of the composites is shown in Figure 5a, which shows that composites with 100 bar compression pressure showed better strength. At this given pressure, suitable crystallinity and compact adherence of the constituents were achieved. Comparatively, 100 bar compression pressure provides an optimum level of crosslinking in the polymer network and with nano-particles. The composite processed at this pressure bore higher stress and elongation, due to a suitable combination of crystalline and amorphous domains. Higher pressure (120 bar) facilitates larger-sized crystalline domains, which reflects the brittle nature of the composite. Hence lower elongation was observed for the composite moulded at 120 bar pressure. At lower pressure (80 bar), mutual adherence of polymer chains among themselves and with nano-particles was less, hence lower strength was observed in a poorly interconnected network. The tensile modulus (initial modulus) of the composites was 21.7, 20.8, and 17.6 GPa (Figure 5b), at a compression pressure of 120, 100, and 80 bar, respectively. Composite with lower compression pressure showed lower initial modulus due to the higher content of amorphous domains. Higher elongation is expected for the higher content of amorphous domains under tensile loads. Whereas, at a compression pressure of 100 bar, a mixed phase of crystalline and amorphous domains exists, which induces better elongation and tensile strength in the composite structure.

Figure 5c shows the flexural modulus of the composites, i.e., 1.76, 1.98, and 2.27 GPa at 80, 100, and 120 bar compression pressure, respectively. The Figure 5d showed that composite fabricated with higher compression pressure offers better flexural strength, due to its higher crystallinity, which increases its resistance to bending. Lower pressure depicted lower crystallinity of the composites, so higher empty spaces render bending of composites at lower stress. Brinell hardness also showed a similar trend to that of flexural resistance, as shown in Figure 5d. Composites showed Brinell hardness values of 69 at 80 bar, 72 at 100 bar, and 76 at 120 bar compression pressure. Hardness increases as crystallinity increases, so the composite with higher compression pressure showed better results in Brinell hardness. Composites fabricated at lower pressures showed comparatively lower Brinell hardness, due to less crystallinity of the composites. Figure 5e shows the impact energy absorption of the composites, which was 18.4, 45.2, and 43.1 J for the composites fabricated at a compression pressure of 80, 100, and 120 bar, respectively. The Figure 5e shows the best impact absorption was observed at 100 bar compression pressure, which is due to the integration of both high strength and elongation in a mixed-phase structure.

In summary, a suitable crystallinity was achieved at a compression pressure of 100 bar, where strength and elongation both are offered in the composite structure. Based on better tensile and impact strength; along with comparable flexural and Brinell hardness, we can choose 100 bar compression as optimum for composite fabrication.

#### 3.2.3. Effect of Reinforcing Agent

Particulate reinforcement increases the mechanical properties of the composite. The density, hardness, compressive strength, and toughness increased with increasing reinforcement fraction; however, these properties may reduce in the presence of porosity in the composite material. In this work, the mechanical properties of the composites are tested against four different concentrations of the reinforcement, i.e., 0, 2, 4, and 6 wt%. UP resin without nano-reinforcement is denoted as 0%, which was taken as a reference sample for comparison. At this stage, we used the curing temperature of 80 °C and compression pressure of 100 bar for all samples. Stress-strain curve of the composites with different content of nano-inclusions is shown in Figure 6a. It is obvious from the tensile curve that the composite containing 4% of nano-reinforcement showed maximum stress and elongation at the break. Higher concentrations of nano-reinforcement (6%) showed inferior results, which are due to agglomeration, improper dispersion, and less adhesion with the matrix. At higher concentrations, the particles tend to drastically suppress the free volume, hence the brittle nature is induced in the sample. Relatively, at a 4% concentration of nano-reinforcement, there is negligible agglomeration and less stiffness of composite; so it can be approximated that a strong interface of nano-particles and the resin exists at 4% concentration. Compared to 0% concentration, the optimized (4%) concentration showed two folds with higher strength and 66% higher elongation at break. Thus, with a small content of room-temperature synthesized nano-inclusions, a significant improvement in the mechanical properties was achieved. Intriguingly, the dispersion of room-temperature-produced TiO_2_ nanoparticles did not need the use of a surfactant or stabilizing agent, leading to a notable improvement in the characteristics of the resin.

A similar trend of enhancement in the tensile modulus, flexural modulus, and impact resistance was recorded for 4% concentration, as shown in Figure 6b–e, respectively. There was a linear increase in the tensile modulus (up to 23 GPa) with an increase in the concentration of nano-reinforcement up to 4% (Figure 6b). In contrast, the PU resin showed a tensile modulus of 14.85 GPa, which is due to a loosely and weakly interconnected network of polymer chains. Similarly, the flexural modulus value of 2.68 GPa was recorded for the 4% nano-inclusion, which is two folds higher than the pure PU resin (Figure 6c). PU resin showed Brinell hardness values of 59, i.e., 0% reinforcement, whereas hardness was 78 at 4% reinforcement (Figure 6d). Based on better strength and elongation of 4% reinforced UP resin, the impact resistance of the composite structure was also expected to be higher. The impact resistance of UP resin was 23.98 J without nano-reinforcement, whereas an impact resistance of 50.97 J was recorded for 4% reinforcement (Figure 6e). The impact strength was doubled with the nano-inclusions, which was related to the strong interface between the nanoparticles and resin. Individual particles dispersed in the resin can be seen in Figure 7a,b, composite showed better adhesion and proper crosslinked structures with 4% TiO_2_ nanoparticles inclusion. While higher inclusion showed aggregation caused voids, improper adhesion, lower strength, and less crosslinking of TiO_2_ nanoparticles with the resin shown in Figure 7c.

In conclusion, we can consider that the optimum concentration of the TiO_2_ nano-inclusion was 4%, where a significant enhancement in the mechanical properties was recorded. Suitable processing temperature, compression pressure, and concentration of nano inclusions result in drastically better performance. The possible mechanism enhanced mechanical properties via nano-reinforcements is through blockage of crack at a very initial stage. Furthermore, due to the superior functionality of the proposed room-temperature synthesized TiO_2_ nanoparticles, a strong interface between resin and nanoparticles exists. These strong interactions essentially provide a better stress distribution, hence mechanical performance is improved. The polymer chains also tend to align themselves on the nano-particle surface; hence the multidimensional orientation of chains induces better performance characteristics.

### 3.3. Functional Characterization

Apart from enhancement in mechanical properties, the antibacterial activity of UP and TiO_2_-UP composite resin is shown in Figure 8. The optimized sample with 4% TiO_2_ inclusions was subjected to the antibacterial test. As TiO_2_ is doped into UP resin, antibacterial activity is enhanced and it hinders the growth of bacteria on its surface. Unlike TiO_2_-coated samples or TiO_2_ powders’ antibacterial activity, a zone of inhibition was not created around the sample shown in Figure 8a. Here, leaching out of the TiO_2_ was not feasible because the TiO_2_ is crosslinked in the polymer network. Bacterial growth was inhibited surrounding the TiO_2_-UP composite sample, yet, there was no sign of inhibitory zone. In contrast, the growth of dense bacterial colonies was observed around and on the edges of the pristine UP resin sample shown in Figure 8b [38]. This antibacterial activity of TiO_2_ is due to its semiconductor nature, which imparts a photocatalytic effect in TiO_2_. Reactive oxygen species are formed on the surface of TiO_2_ due to the photocatalytic effect, which readily degrades the bacteria’s membrane, due to their reactive nature [38].

Table 1 compares the performance of room-temperature synthesized nano-particles with the conventional high-temperature synthesized nano-particles. Due to the limited functionality of such particles, maximum of 2% doping concentration can be dispersed in the polymer resins. In contrast, the small size and highly functional surface of the room temperature synthesized TiO_2_ enables it to be dispersed up to 4%. In addition, the strong interface provides better stress distribution, due to which our proposed composite enhanced the mechanical strength by two folds. Based on the drastically better performance and scalability of the process, we expect that the proposed method can advance the application of nano-composites at a commercial level.

## 4. Conclusions and Future Work

In this study, a facile method for the scalable synthesis of TiO_2_ nanoparticles was used, for the enhancement of the mechanical properties of the UP matrix. Nanoparticles showed a spherical morphology, with an average size of 50–80 nm. Highly functional surface and crystalline structure of particles were observed in FTIR and XRD, respectively. In addition, the processing conditions of the composite structure were optimized, including curing temperature (80 °C), compression pressure (100 bar), and concentration (4%) of nanoparticles. The optimized reinforced sample showed nearly two folds higher mechanical properties, as compared to the pristine UP matrix. In addition, suppressed bacterial growth was observed on the TiO_2_-UP composite structure. The superior mechanical performance of the composite was related to the effective multidimensional stress distribution and blockage of crack generation/propagation at a much smaller scale. This significant improvement in mechanical performance, with a low concentration (4%) of TiO_2_ nano-inclusions holds great promise for the industrial-scale application of nano-composites. Also, such bioactive composites are essential for hospital door handles, which can be a potential source of disease spread. These composites also found application in the biomedical field such as prosthetics and orthopedics for bone fixation plates, hip joint replacements, bone cement, and bone transplant.

Room-temperature synthesized nano-flowers and hollow spheres can be used for nanocomposite applications in the future. In addition, the surface modification of fibers used in composite reinforcement can be carried out via in situ room temperature synthesis of nanparticles on fibrous substrates.

## Figures and Tables

**Figure 1 polymers-15-00934-f001:**
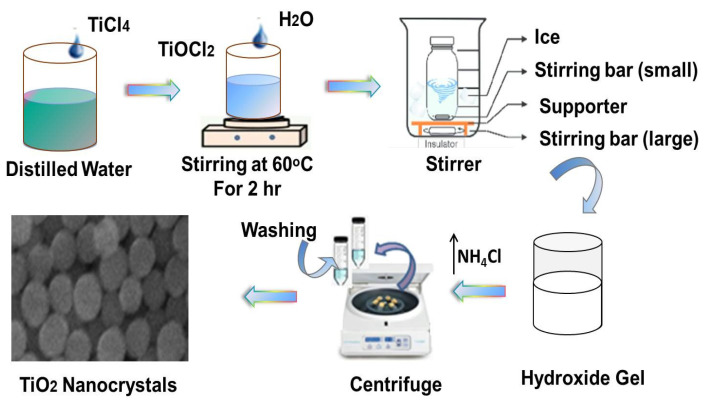
Schematic illustration of the TiO_2_ nanoparticles synthesis process.

**Figure 2 polymers-15-00934-f002:**
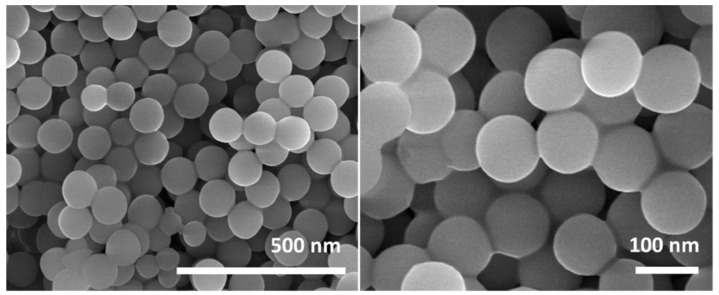
SEM images of the TiO_2_ nanoparticles at different resolutions.

**Figure 3 polymers-15-00934-f003:**
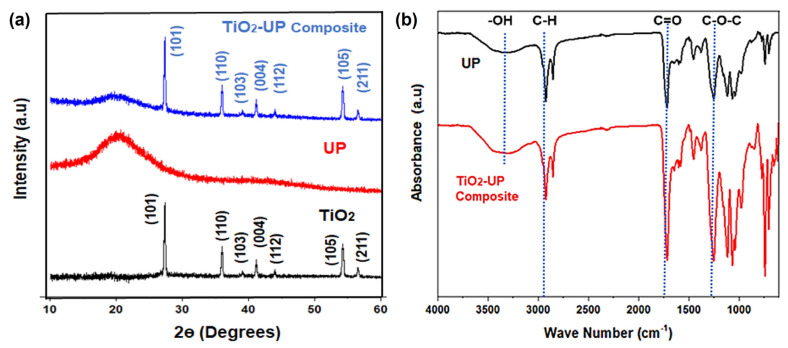
Comparison of (**a**) XRD Spectra and (**b**) FTIR spectra of unsaturated polyester (UP) resin and TiO_2_−UP composite.

**Figure 4 polymers-15-00934-f004:**
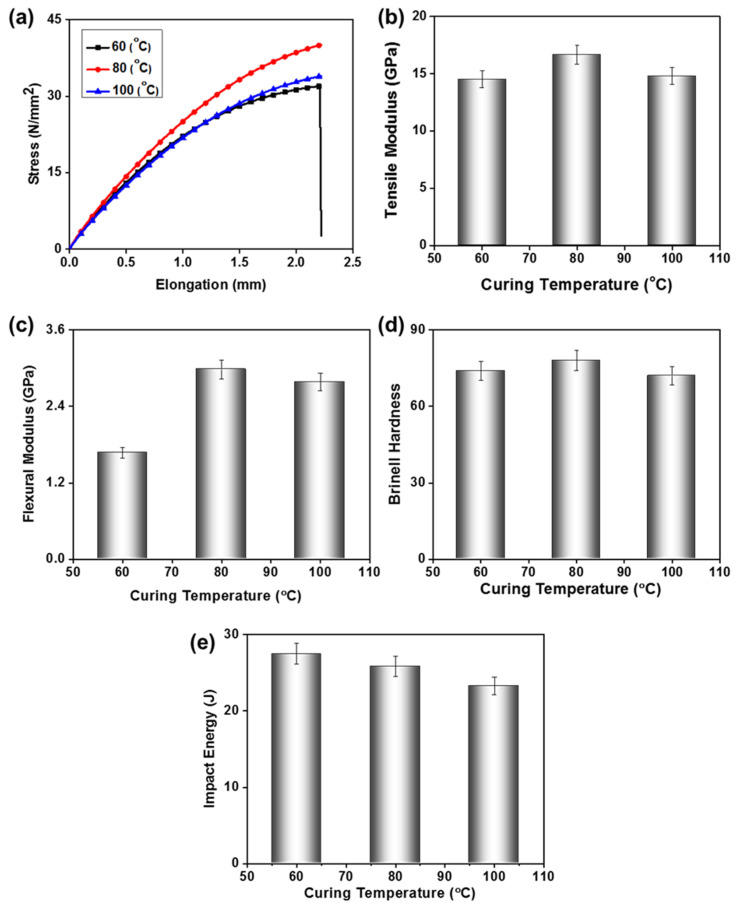
Effect of curing temperature on (**a**) tensile strength, (**b**) tensile modulus, (**c**) flexural strength, (**d**) Brinell hardness, and (**e**) impact resistance of TiO_2_-UP nanocomposite.

**Figure 5 polymers-15-00934-f005:**
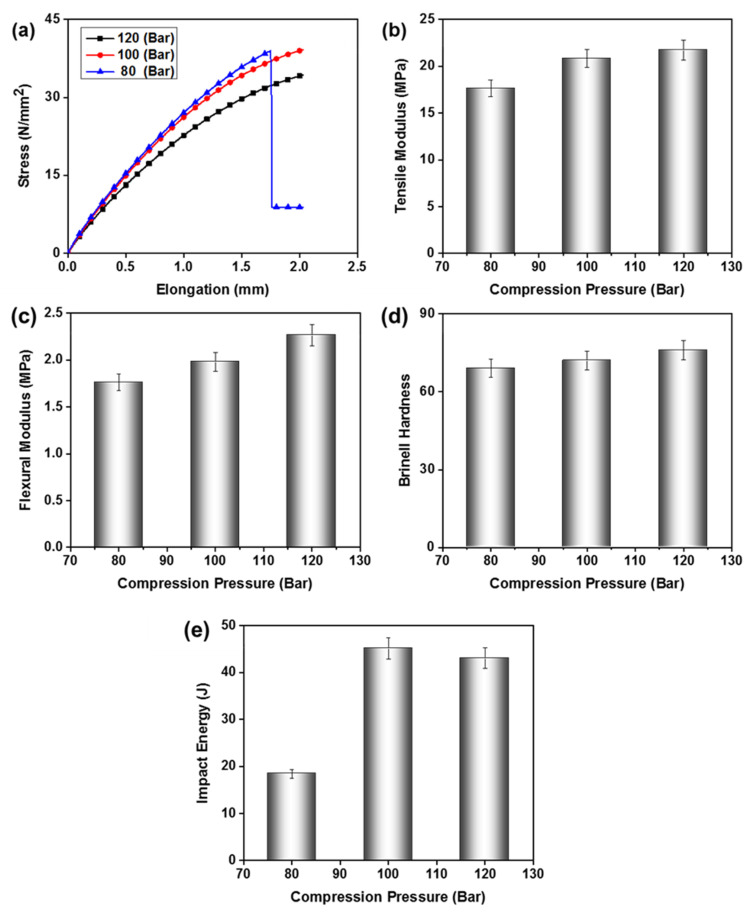
Effect of compression pressure on (**a**) tensile strength, (**b**) tensile modulus, (**c**) flexural strength, (**d**) Brinell hardness, and (**e**) impact resistance of TiO_2_-UP nanocomposite.

**Figure 6 polymers-15-00934-f006:**
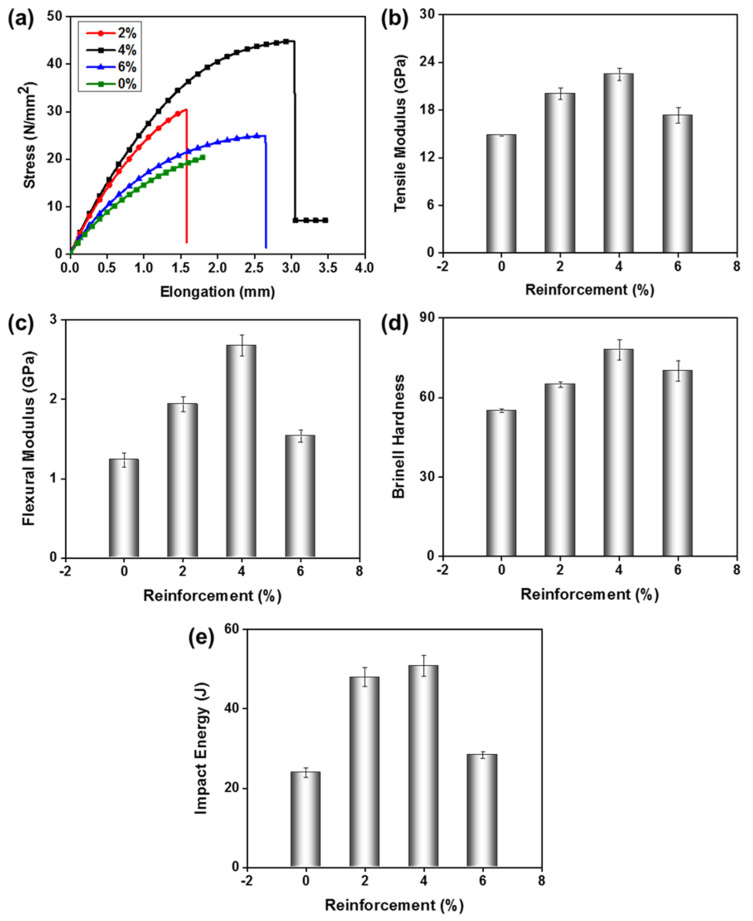
Effect of concentration of nano-reinforcement on (**a**) tensile strength, (**b**) tensile modulus, (**c**) flexural strength, (**d**) Brinell hardness, and (**e**) impact resistance of TiO_2_−UP nanocomposite.

**Figure 7 polymers-15-00934-f007:**
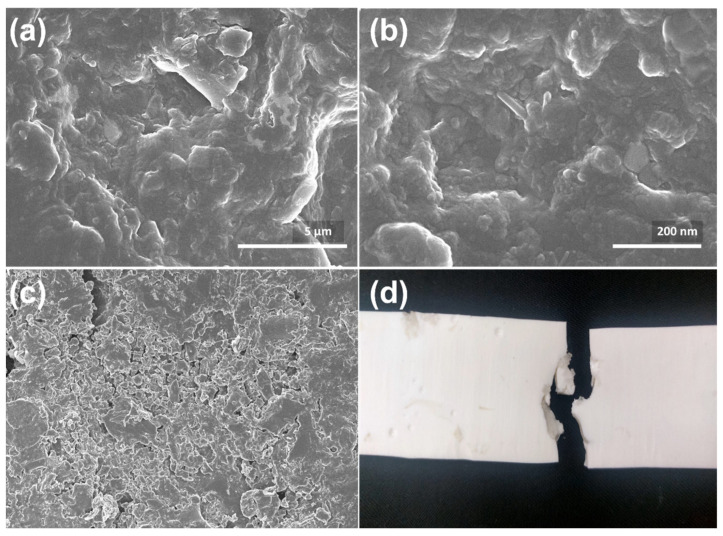
SEM images of the (**a**,**b**) optimized composites that showed proper adhesion and cross-linking of TiO_2_ nanoparticles, (**c**) while aggregation caused voids and improper adhesion that caused lower strength. (**d**) Fracture image of the TiO_2_-UP composite.

**Figure 8 polymers-15-00934-f008:**
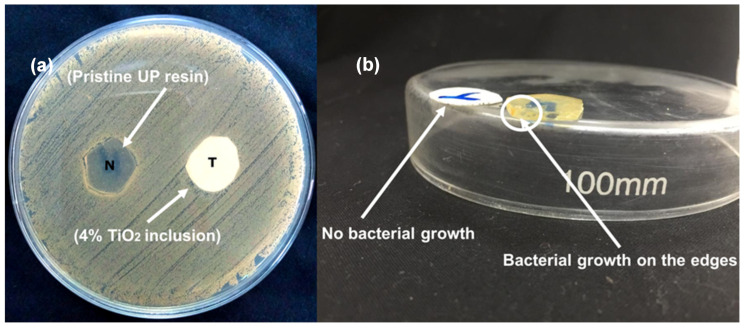
(**a**) Comparative antibacterial efficacy, bacterial growth is seen around the pristine UP resin sample and edges, (**b**) 4% TiO_2_-UP has showed no growth on surface.

**Table 1 polymers-15-00934-t001:** Comparison of proposed work with previously reported nano-fillers.

Material	Maximum Dispersable Concentration	Synthesis Method	Mechanical & Chemical Properties	Additional Functionality	Ref.
**SiO_2_**	2%	Hydrothermal synthesis	20.61% in tensile strength, 23.71% in compressive strength, and 22.88% in impact strength were attained	NA	[2]
**SiO_2_**	2%	Hydrothermal	Improve the interfacial compatibility between reinforcement and matrix, 55% more tensile strength with 2 wt% doping	UV blocking, photostability	[40]
**TiO_2_**	0.5%	Hydrothermal synthesis	Improve mechanical performance under various stresses (storage modulus, loss modulus, tensile strength, and modulus)	-	[29]
**TiO_2_**	0.7%	Hydrothermal synthesis	The increase in tensile, flexural, impact, and interlaminar shear strength values were observed as 10.95%, 20.05%, 10.45%, and 18.80%, respectively.	The water diffusion coefficient was reduced by 31.66%	[28]
**TiO_2_**	4%	Room temperature synthesis	Two folds enhanced tensile strength and 66% higher elongation at break	Antibacterial activity was enhanced	This work

## Data Availability

Not applicable.

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
