# Peer review of "Room Temperature Synthesized TiO2 Nanoparticles for Two-Folds Enhanced Mechanical Properties of Unsaturated Polyester"

_polymers, 2023, doi:10.3390/polym15040934_

Round 1

Reviewer 1 Report

In this article, the authors introduce a facile method for TiO2 nanoparticles and use it for enhancing the mechanical properties of unsaturated polyester. The article has a clear idea, clear frame, and logical language; thus, I enthusiastically support acceptance of this work after addressing the following issues.

(1) The introduction is not written logically enough, and the background is not expressed through the introduction. While it addresses the main issues addressed by the work, it is in a confusing logical order, such as the micro filler introduced at the beginning of the second paragraph, which is not the main idea in the article.

(2) In Figure 1, the last step is the centrifuge and wash with distilled water, which should be noted in this figure.

(3) We recommend that the scale bar in the bottom image change to the nm level in Figure 2.

(4) We recommend that the articles published in Advanced Powder materials, Advanced Fiber Materials, and Chinese Chemical Letters journals be cited.

(5) An XRD pattern of the prepared TiO2 should be included in Figure 3 (a), which will be evident in your analysis in lines 219-220.

Reviewer 2 Report

I have several comments and questions about the submitted work:

1.       Line 9 – “Use of nano-inclusions to…” – Using of nano-inclusions to…

2.       Line 13 - “… and economic synthesis …” – “and economical synthesis”

3.       Lines 52-53 – What does the sentence “Various nanofillers including CaCO3 [12], SiO2 [2], ZnO [13], and TiO2 [14].” mean?

4.       Line 76 – For what applications is the polymer composite with TIO designed?

5.       Line 96 – “(TiCl4, 94 %)” it should be “(TiCl4, 94%)”, like it is at H2O2 and in the line 99

6.       Line 122 – “Schematic illustration for the synthesis process of TiO2 nanoparticles.” Please consider the change of Fig. capture: Schematic illustration of the TiO2 nanoparticles synthesis process.

7.       How many grams of TiO2 did the authors receive using 5.5 ml of TiCl4 and 200 ml of distilled water?

8.       Line 130 – What does this sentence say? “The mass of the filler material was kept 30% of the weight fraction of the polyester resin, and the mass of the polyester resin was calculated according to the required volume of the cast.”

9.       Figure 2 – There is no need to show three SEM photographs.

10.   Why do the authors decide to increase the strength of the polymer using Tio2? Aren't other compounds that are cheaper and more suitable?

11.   Line 128 – Authors are writing: “The effect of concentration of reinforcing agent TiO2 is by varying its quantity, i.e. 5%, 10%, and 15%.”  So, 5%, 10%, and 15% of TiO2 were used. But in line 257, there the composite with 2% of TiO2 is tested. Why?

And again, in line 340 there are different amounts 0, 2, 4, and 6 wt.%. ????

12.   It would be appropriate to show the samples after testing, and what the materials and the fractures look like.

13.   “Higher concentrations of nano-reinforcement (10%) showed inferior results, which are due to agglomeration, improper dispersion, and less adhesion with the matrix.” There are no results of 10% reinforcement.

14.   Chapter” Effect of reinforcing agent - If the authors applied more intensive homogenization of Tio2 particles, it would be possible that even higher quantities would achieve better distribution and thus mechanical properties? Perhaps TIO agglomeration at 6% is caused by an inappropriate preparation procedure.

15.   The authors should show SEM images of the distribution of particles in the matrix (for all TiO2 contents). Otherwise, it is not possible to talk about agglomerations or poor particle distributions.

16.   The results of antibacterial activity must be redesigned. The figure is not clear, no inhibition zone is observed but the authors claim that the material has antibacterial properties. The chapter is unclear, the results should be supported by photos with inhibitory zones (pure polymer, TiO2 particles, and composites with different TiO2 content).

I ask the authors to mark all the repairs they do in the text.

Only after answering all the questions, I can recommend this work for publication.

Round 2

Reviewer 2 Report

Thanks to the authors for accepting the comments. I have no further comments on the work.